# Fully Dynamic High–Resolution Model for Dispersion of Icelandic Airborne Mineral Dust

Bojan Cvetkovic [1,*], Pavla Dagsson-Waldhauserová [2,3], Slavko Petkovic [1], Ólafur Arnalds [2], Fabio Madonna [4], Emmanouil Proestakis [5], Antonis Gkikas [5], Ana Vukovic Vimic [6], Goran Pejanovic [1], Marco Rosoldi [4], Darius Ceburnis [7], Vassilis Amiridis [5], Lenka Lisá [8,9], Slobodan Nickovic [1,10] and Jugoslav Nikolic [1]

[1] Republic Hydrometeorological Service of Serbia, Department of National Centre for Climate Change, Kneza Viseslava 66, 11000 Belgrade, Serbia
[2] Faculty of Environmental and Forest Sciences, Agricultural University of Iceland, 311 Hvanneyri, Iceland
[3] Faculty of Environmental Sciences, Czech University of Life Sciences, 165 21 Prague, Czech Republic
[4] Consiglio Nazionale Delle Ricerche, Istituto di Metodologie per l'Analisi Ambientale, 85050 Potenza, Italy
[5] Institute for Astronomy, Astrophysics, Space Applications and Remote Sensing, National Observatory of Athens, 15236 Athens, Greece
[6] Faculty of Agriculture, University of Belgrade, Nemanjina 6, 11080 Belgrade, Serbia
[7] Ryan Institute Centre for Climate and Air Pollution Studies, National University of Ireland, H91 CF50 Galway, Ireland
[8] Institute of Geology, Czech Academy of Sciences, 165 00 Prague, Czech Republic
[9] CRL Radiocarbon Laboratory, Nuclear Physics Institute of the Czech Academy of Sciences, Na Truhlářce 39/64, 180 86 Prague 6, Czech Republic
[10] Institute of Physics Belgrade, University of Belgrade, Pregrevica 118, 11080 Belgrade, Serbia
* Correspondence: bojan.cvetkovic@hidmet.gov.rs

**Abstract:** Icelandic topsoil sediments, as confirmed by numerous scientific studies, represent the largest and the most important European source of mineral dust. Strong winds, connected with the intensive cyclonic circulation in the North Atlantic, induce intense emissions of mineral dust from local sources all year and carry away these fine aerosol particles for thousands of kilometers. Various impacts of airborne mineral dust particles on local air quality, human health, transportation, climate and marine ecosystems motivated us to design a fully dynamic coupled atmosphere–dust numerical modelling system in order to simulate, predict and quantify the Icelandic mineral dust process including: local measurements and source specification over Iceland. In this study, we used the Dust Regional Atmospheric Model (DREAM) with improved Icelandic high resolution dust source specification and implemented spatially variable particle size distribution, variable snow cover and soil wetness. Three case studies of intense short- and long-range transport were selected to evaluate the model performance. Results demonstrated the model's capability to forecast major transport features, such as timing, and horizontal and vertical distribution of the processes. This modelling system can be used as an operational forecasting system, but also as a reliable tool for assessing climate and environmental Icelandic dust impacts.

**Keywords:** high latitude dust; coupled atmospheric–dust model; dust sources; Icelandic soils and sediments; volcanic glass

## 1. Introduction

Currently, there is no operational dust modelling system in the community designed to predict Icelandic airborne dust processes. Namely, horizontal resolutions of current operational dust models do not adequately represent the spatial distribution of Icelandic dust sources, especially small-scale but highly emissive areas defined as "hot-spots". A regional atmospheric–dust model, if applied with horizontal grid size of several kilometers, can identify such sources and their physical characteristics and could consequently reproduce

dust concentration patterns. Several studies based on regional modelling have been used to simulate dust transport from Iceland. Many of them are based on atmospheric weather prediction systems used to "offline" drive the dust process by implementing Lagrangian models such as HYSPLIT [1], FLEXDUST [2] or NAME [3]. The major conceptual and practical constraint of such predictive systems is the lack of simultaneous interactions between the meteorological parameters and the dust concentration during the model simulation/forecasting.

Icelandic deserts and glacial sediment areas constitute the largest European source of airborne mineral dust [4,5]. Iceland is an active volcanic island where many of the volcanic systems are capped by glaciers, which results in frequent explosive volcanism [6]. Glacial rivers have a high load of basaltic glass fragments, and many of the main dust sources are associated with glacial margins, glacial rivers, and sandy ocean beaches near glacio-fluvial outlets [4]. The soils that form in the basaltic dust deposits are andosols. By lacking cohesive forces, they are extremely susceptible to wind erosion and to dust emission. The soil particles are primarily volcanic glass and basalt fragments together with allophane (commonly 5–15%) and ferrihydrite clay (commonly 3–10%), depending on the degree of weathering, with lower clay content at the dust hot-spots, but there are no phyllosilicates such as smectite [7]. Dust release from periglacial sources is well documented, such as for Alaska and Iceland, and may be the major source of dust in high latitudes. Dust release may likely be increasing with retreating glaciers due to global warming [2,5,8–10].

Soil geochemistry in dust sources in Iceland show high iron content, usually about 10% Fe, which is much higher than in continental dust in general [11–13]. The iron is an essential micronutrient for marine microbial organisms and is an important modulator of $CO_2$ uptake from the atmosphere in the high latitude North Atlantic [14], thus contributing to the slowdown of the ongoing Arctic seawater acidification. In recent years, the high latitude North Atlantic Ocean has become a focus for research into the role of Fe in ocean productivity [15].

Estimated mean annual dust emissions from Icelandic desert surfaces are of the order of 5 to 40 million tons, with the majority of the dust deposited on land or near the shore-lines [2,12]. However, dust particles from Iceland can be transported at long distances [4], and they have been identified more than 3500 km far from the sources [16–21]. There are, on average, 135 dust days annually observed at weather stations in Iceland [22–24], but occurrence frequency of dust emission and transport is much higher, as some of the dust is blown from beach areas directly out to sea. Dust day frequency in Iceland is higher than in USA, Mongolia, Iran, as well as Saharan dust day occurrence in the Mediterranean basin [25]. Dust storms in the highlands occur mainly in summer (May–October) because several of the main dust sources, such as Dyngjusandur, Mælifellsandur and Hagavatn, are usually covered with snow during winter months [23,26]. The Arctic dust storms in NE Iceland directly affect Arctic areas. In [2], it was estimated that about 7% of Icelandic dust reaches the High Arctic (>80° N). Additionally, [20,27,28] identified that dust pathways reach far north. Deserts in South Iceland are dust productive continuously throughout the year, with maximum frequency in spring (March–May). The frequent appearance of dust storms is related to high velocity wind conditions typical for high latitude regions [29]. Finally, there is evidence that some dust storms occur during a vast variety of weather conditions [22,30,31]. Most of the dust storms affect Iceland and its neighborhood, but in some cases dust can reach distant regions such as the Faroe and British Islands, traveling mainly in the lower troposphere.

Icelandic dust storms with concentrations often exceeding 1000 µg m$^{-3}$ reduce air quality, causing low visibility in urban areas [1,22,26,30,32]. Icelandic road transportation is strongly affected by dust storms more than 20 times per year on average, resulting in road closures due to reduced visibility [33,34]. High $PM_{10}$ pollution levels in Reykjavik tend to be significantly associated with emergency hospital visits and increased need for anti-asthmatic drugs in the adult population [35,36]. The contribution of submicron dust particles to measured $PM_{10}$ during dust storms in Iceland can exceed 50% ($PM_1/PM_{10}$

ratio of 0.5), which is comparable to urban air pollution rather than to natural dust air pollution elsewhere [32]. Icelandic volcanic dust can affect high latitude climate through both direct and indirect forcing. Being dark in color with spectral reflectance of near black body of <0.03 under laboratory conditions, it can reduce snow albedo similarly as black carbon [37–40]. Furthermore, it is a radiatively active aerosol that can absorb both short and long-wave radiation and can also impact the atmospheric stability by altering radiation budget [41–43]. The net effect is represented by warming of the top of the dust layer and cooling of the surface, resulting in more stable planetary boundary layer. This can additionally enhance the health risk by prolonging the high near surface air pollution. Finally, Icelandic dust is assumed as an important source of ice nucleation particles [44], which impacts climate through indirect aerosol effects.

The aim of this study is to present for the first time a fully dynamic high-resolution numerical atmospheric–dust modelling system, capable to simulate/forecast the atmospheric cycle of mineral dust emitted from the Icelandic soil sources. To understand and mathematically represent the mechanism of airborne dust, processes ranging from micro to global scales, which include dust emission, horizontal and vertical turbulent mixing, long-range transport and dust deposition, have to be considered in order to reflect the complexity of the atmospheric dust cycle.

## 2. Materials and Methods

The data used in this study include: field campaigns data used to define soil characteristics, required as model input data; atmospheric dust transport model results obtained in numerical simulation of the dust storm cases; in situ and satellite observed aerosol data as well as air quality observations used to define main characteristics of the dust events and to verify model results.

### 2.1. Numerical Dynamic Modelling of Icelandic Dust Atmospheric Process

In order to overcome the current limitations in the Icelandic dust modelling, we introduce here for the first time a high-resolution numerical coupled atmospheric-dust model with detailed representation of the Icelandic soil sources. The model is designed to perform the missing daily dust predictions, but also to simulate diverse impacts of this dust to various environmental systems. For this purpose, we implement the Eulerian-type numerical, fully coupled, atmospheric–dust model DREAM (Dust REgional Atmospheric Model) [45,46] and simulate the Icelandic dust atmospheric process, from its emission to deposition. Particular effort has been made to include information on emission from Icelandic fine-scale dust sources. Finally, model performance has been evaluated using in situ and satellite observations for specific cases characterized by significant dust emission and long-range dust transport.

### 2.2. Parameterization of Dust Emission from Icelandic Sources

Geographical distribution of dust sources is a key input information in numerical atmospheric–dust models. Emission of dust particles from the surface is generally determined by the soil surface conditions (soil moisture and temperature, soil texture, looseness of the soil surface, land cover characteristics) and by near-surface weather conditions. Dust sources are bare fraction of topsoil or sediment surfaces susceptible to wind erosion. Favorable conditions for emission of soil particles from such surfaces include: lower topsoil moisture, unfrozen soil and surface wind velocity above certain threshold closely related to soil particle size distribution and soil moisture. Soil surfaces are susceptible to wind erosion when they contain clay- and/or silt-sized soil particles. Dust sources can be permanent, seasonal, or just appear occasionally in extreme weather conditions. Their dynamics in activity can be impacted by the weather and human activities, predominantly by agriculture practice and water scarcity [47,48]. Volcaniclastic sandy deserts in Iceland cover about 22,000 km$^2$, but most of the material suspended during the dust storms comes from seven major dust sources "hot-spots" according to [4,7,49]. These hot-spots are mainly located on

glacio-fluvial plains and they cover about 500 km$^2$. Dust from these areas of silty sediments has particle diameters of less than 30 μm [50]. Icelandic dust particles contain amorphous glass, large internal voids, and dustcoats comprised of nano-scale flakes. They have high porosity and roughness while particle densities and settling velocities are low.

To define geo-referenced data on the Icelandic dust sources (Figure 2), we use the following high-resolution digital datasets:

(A) ***Desert geomorphology*** with the resolution of 1:100,000 is based on the survey of soil erosion in Iceland [51]. Soil surfaces, according to vulnerability to erosion, are classified in: ***extreme***, ***severe*** and ***considerable*** exposure to erosion (Figure 1, colored areas). These areas indicate possible potential for mineral dust emission in case of suitable weather, topsoil and land cover conditions. The dataset also includes the geographic locations of the glaciers at the time of dataset publication.

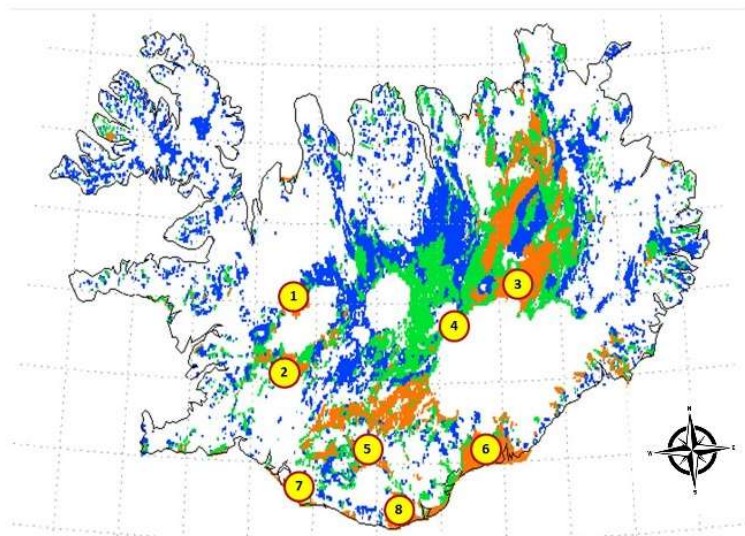

**Figure 1.** Areas vulnerable to erosion (extreme—orange, severe—green, considerable—blue) and hot–spots of dust emission (yellow circles); Dust hot-spots geographical names [1—Flosaskarð, 2—Hagavatn, 3—Dyngjusandur, 4—Vonarskarð, 5—Mælifellssandur, 6—Skeiðarársandur, 7—Landeyjarsandur, 8—Mýrdalssandur].

(B) ***Dust hot-spots*** with the resolution of 1:15,000 represent small-scale dust source patterns with a potential emissivity that far exceeds dust productivity of the sources from (A), according to [4] (Figure 1, marked with numerated yellow circles).

The geo-referenced dust source spatial distribution map for Iceland is made by combining (A) and (B) data sets into a unique dust mask with values ranging from 0 to 1 (Figure 2), where values are proportional to dust emission potential, with the highest values corresponding to highly emissive sources. In accordance with the up-to-date most reliable observational data, the highest weight is given to detect hot-spot source areas and to highlight contributions of these sources, which are noted to be responsible for selected dust transport cases. Other erodible surfaces are included as sources with smaller contributions. These surfaces encompass all potentially dust productive areas capable to produce at least "blowing dust" events [52].

Thereby, the contribution of soil surfaces to potential dust emission are weighted with 0.75 for hot-spots (confirmed dust emissive areas) and 0.25 for the other erodible surfaces. Vulnerability to erosion from these other surfaces, categorized as extreme, severe and considerable (erosion classes 5, 4 and 3 in [51]), are weighted by 0.2, 0.04 and 0.02, respectively.

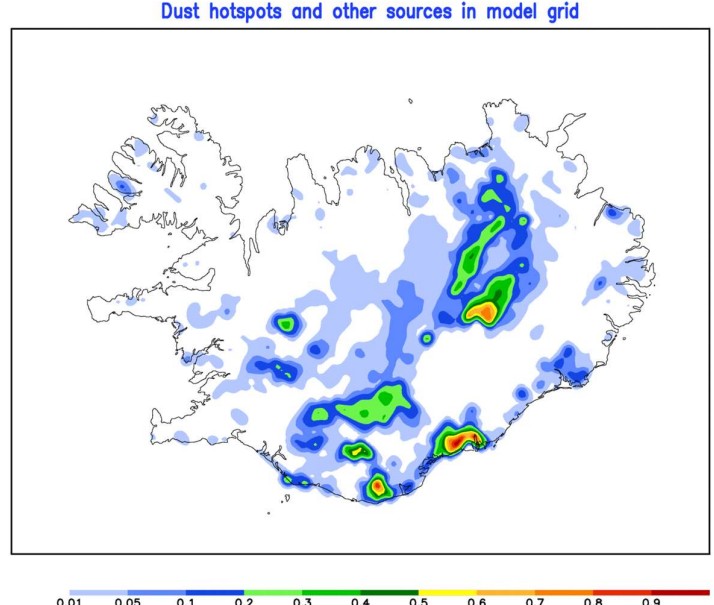

**Figure 2.** Derived dust source mask for Iceland as seen on the model horizontal resolution of 0.05 degrees.

An important soil feature that influences dust emission is the soil texture. Soil surface can be susceptible to wind erosion if it contains clay and silt size particles—up to about 50–60 μm in diameter. During the extreme weather with high surface wind velocity, larger particles can also be suspended from the surface, but over much shorter distances than silt and clay particles. In DREAM, clay and silt particles are included to represent the airborne dust life cycle. Clay particles exist mostly as silt- and sand-sized aggregates and because they are exposed to stronger adhesive forces, they are generally much less available for emission than silt-sized particles [53]. Clay is not a prominent fraction of the Icelandic desert surfaces. Higher clay content is only for the developed andosols, which are not dominant emission sources in the model. In this study, we use the soil texture geographical distribution with the resolution of 1:100,000 [51,54] to populate their fractions over the model grid points, as shown in Figure 3. The clay fraction in model grid is shown in Figure 3a. Higher silt contents (Figure 3b) are typically due to the glaciogenic origin of the sediments, making up the major proportion of sediment loads of glacial rivers. In this dataset, fine particles content in identified hot-spot areas is low [55] (Agricultural University of Iceland, field collected sample data). Accordingly, fine particles content in hot-spots is updated using data from the same field campaign.

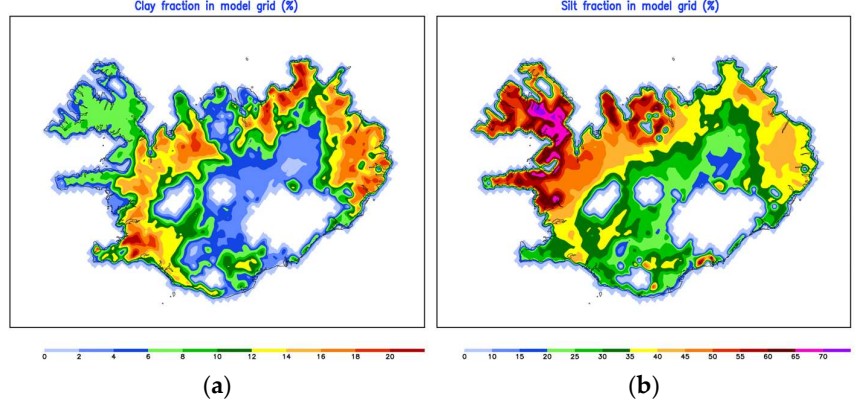

**Figure 3.** Geographic distribution of clay and silt fractions mapped to the model grid with the resolution of 0.05 degree; (**a**) clay fraction; and (**b**) silt fraction.

The total mass of the emitted dust and its gravitational settling in the process of removal from the atmosphere strongly depends on the particle size distribution at dust sources. Most of the current dust models use a particle size constant with respect to geographical distribution [56–58]. In Iceland soils, particle size distributions measured at eight hot spot locations indicates that particles subjected to dust emission can be larger in size than, e.g., particles from the Saharan desert as the Icelandic sources have not been exposed to long-term sorting of particles by aeolian processes. Unlike what is done in other dust models, we incorporated the data obtained at the sites as location-dependent information at hot spots. For productive dust sources other than hot-spots, we use the arithmetic mean of size distributions at hot spots. In the model, we specify 8 particle bins: with radii of 0.18, 0.23, 0.38 and 0.73 μm for clay particles, and with radii of 1.5, 3, 6 and 9 μm for silt particles. Figure 4 shows the normalized particle size distribution at hot spots (Appendix A), along with their arithmetic mean (purple line).

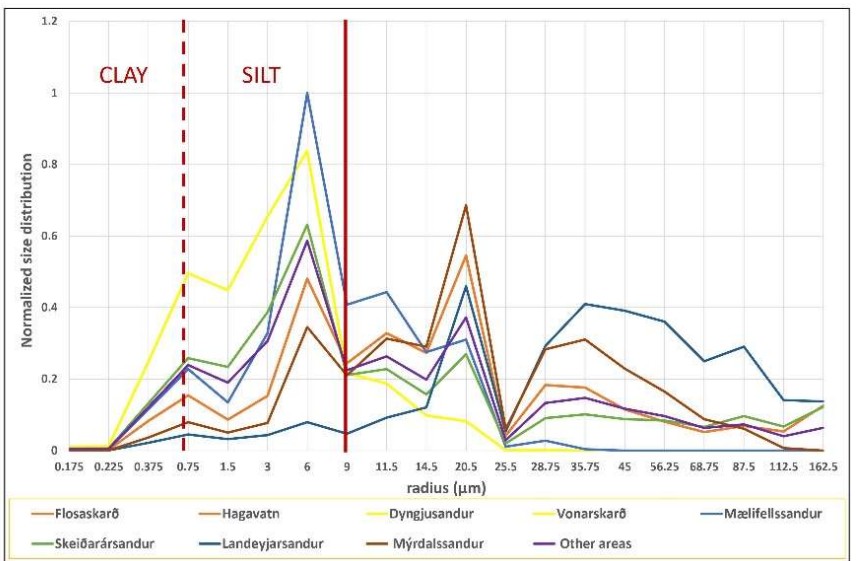

**Figure 4.** Normalized model particle size distributions at measurement sites. Clay and silt particle size domains are indicated with vertical dark red lines. The distributions at Flosaskarð and Hagavatn are the same. Dyngjusandur and Vonarskarð have the same distributions as well.

The near surface concentration $C_s$ of emitted particles, under wind conditions strong enough to create emission, is parameterized in the model as shown in Equation (1):

$$C_s = \alpha \times u_*^2 \left[ 1 - \left( \frac{U_{*t}}{u_*} \right)^2 \right] \; for \; u_* > U_{*t}, \tag{1}$$

where $\alpha$ represents a tuning constant, $u_*$ is friction velocity [44]. The threshold friction velocity $U_{*t}$ is the minimum friction velocity required for triggering dust emission and strongly dependent on the soil wetness and particle size. Therefore, high soil moisture reduces the dust emission. We apply the parameterization from [59], which links dust emission with the soil moisture content (*SMC*), given by Equation (2):

$$U_{*t} = u_{*t} \left[ 1 + 1.21 \left( SMC - SMC' \right)^{0.68} \right]^{0.5} \; for \; SMC > SMC' \tag{2}$$

where

$$SMC' = 0.0014(\%CLAY) + 0.17(\%SILT), \tag{3}$$

Otherwise

$$U_{*t} = u_{*t}, \tag{4}$$

In addition, the threshold friction velocity for dry soil depends on soil particle sizes:

$$u_{*t} = A_i \left( 2gr_i \frac{\rho - \rho_a}{\rho_a} \right)^{\frac{1}{2}}, \tag{5}$$

where $A_i$ is empirical-based parameter [45,60] for i-th particle size bin; $g$ is gravitational acceleration; $r_i$ is bin radius; and $\rho$ and $\rho_a$ are particle and air densities, respectively.

Since DREAM is a coupled atmospheric–dust model, it also includes snow cover and precipitation prediction, which alter soil surface conditions and impact emission from dust sources.

### 2.3. Model Dynamics of Dust Transport

As mentioned before, the DREAM model [45,47,61,62] was developed as a fully coupled atmospheric–dust model. In this version, its atmospheric component is NMME (**N**on hydrostatic **M**esoscale **M**odel on **E**–grid) developed by National Centers for Environmental Predictions (NCEP) [63]. The dust component is embedded as one of the governing prognostic equations that solve a set of dust mass continuity equations for eight particle size bins for Icelandic soils with radii ranging in the interval 0.18–9 μm. The first four bins are considered as clay particles, and another four as silt particles.

DREAM simulates all major components of the atmospheric dust processes such as emission, horizontal and vertical turbulent mixing, free atmosphere transport and dust wet and dry deposition. It numerically solves the Euler-based dust mass conservation equation applied for each dust aerosol bin:

$$\frac{\partial C}{\partial t} + u\frac{\partial C}{\partial x} + v\frac{\partial C}{\partial y} + \nabla_h(K_H\nabla_h C) + \frac{\partial}{\partial z}\left(K_V\frac{\partial C}{\partial z}\right) + (w - v_g)\frac{\partial C}{\partial z} - \left(\frac{\partial C}{\partial t}\right)_{SR} + \left(\frac{\partial C}{\partial t}\right)_{SN} = 0, \tag{6}$$

Here, $C$ is the dust concentration; $u$, $v$ and $w$ are horizontal and vertical wind velocity components; $v_g$ is the dust gravitational settling velocity; $\nabla_h$ is the horizontal gradient operator; $K_H$ and $K_V$ are the lateral and vertical mixing coefficients; subscripts $SR$ and $SN$ refer to dust sources and sinks, respectively. During the model integration, dust emission is calculated over model grid-point cells declared as dust potential sources. Once emitted into the atmosphere, the dust aerosol is driven by turbulent vertical mixing, by horizontal and vertical advection and by deposition processes.

The parameterization of the dust source term is based on the approach of mass emissions from the surface–atmosphere interface. The emission is dependent on the near-surface turbulent state and its intensity is regulated by a thin viscous sub-layer (VSL) inserted between the model surface and the first model layer [64]. Within VSL, there is a mixture of turbulent and laminar mixing determined by the intensity of the near-surface wind. Above VSL, the vertical mixing is parameterized by the Mellor–Yamada–Janjic turbulence scheme [64].

VSL operates over the following three different regimes dependent on intensity of the friction velocity: (1) smooth and transitional, (2) rough, and (3) very rough (Figure 5). When the friction velocity exceeds $u_{*r} = 0.3$ ms$^{-1}$, the smooth and transitional regime stops to operate and the flow becomes rough. Dust emission under rough turbulent regime continues until the next velocity threshold $u_{*t} = 0.7$ ms$^{-1}$ is achieved. At this point, the regime switches to very rough turbulence when VSL is completely ceased and the emission becomes fully driven by turbulence [65]. Schematic description of the viscous sub-layer concept is shown in Figure 5.

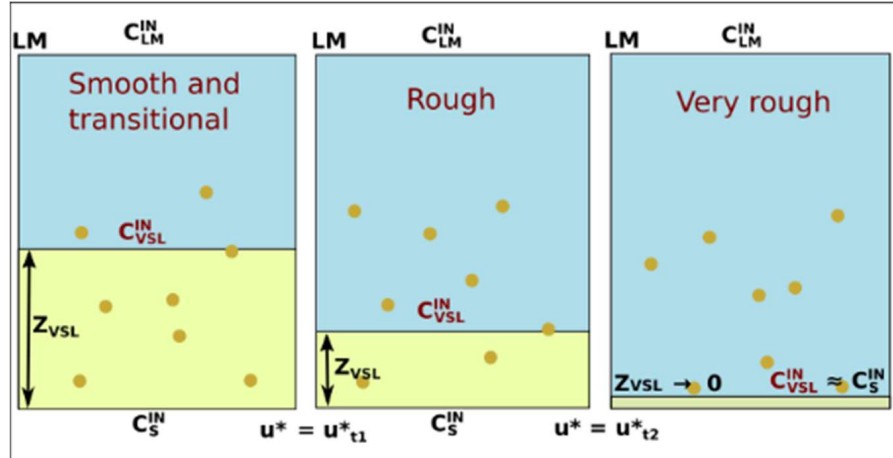

**Figure 5.** Schematic description of the viscous sub–layer concept. Emission under smooth and transitional regime happens for small $u_*$ when VSL is the thickest. In the rough mixing regime, the near-surface turbulence increases, and the VSL depth decreases. Under fully developed turbulent conditions (very rough regime), emission reaches its maximum, and VSL depth vanishes.

The concentration emission flux is expressed in terms of the viscous sub-layer parameters by

$$F_S = v \frac{C_{VSL}^{IN} - C_S^{IN}}{Z_{VSL}}, \tag{7}$$

Here, $v$ is the air kinematic viscosity, and $C_S^{IN}$ and $C_{VSL}^{IN}$ are concentrations at the surface and at the top of VSL, respectively. The lower concentration boundary condition in the model is:

$$C_{VSL}^{IN} = v \frac{C_S^{IN} - \omega C_{LM}^{IN}}{1 + \omega}, \tag{8}$$

Which is a weighted mean of $C_S^{IN}$ and the concentration at the lowest model level $C_{LM}^{IN}$. The weighting factor $\omega$ depends on turbulent and laminar mixing features. $C_S^{IN}$ is a power function of the friction velocity $u_*$ and of its threshold value $u_{*t}$, above which emission begins:

$$C_S^{IN} \sim u^2 \left[ 1 - \left( \frac{u_{*t}}{u_*} \right)^2 \right] for \ u_* > u_{*t}, \tag{9}$$

The dry deposition of dust includes parameterization of gravitational settling, Brownian and turbulent deposition at the air–surface interface, and interception and impaction at the surface roughness elements [45,66]. Deposition scheme takes into account properties of the depositing particles (size, density), features of the depositing surfaces (roughness, land cover, land texture) and turbulent conditions of the lower atmosphere. Different parameterizations are used for the following groups of surfaces: (a) bare soil, ice and sea, and (b) land covered by vegetation. Wet removal of the concentration by precipitation is predicted by the atmospheric model component where, at each model time step, the removal is calculated using a washout parameter [47]. The wet dust removal is proportional to rainfall rate.

### 2.4. Model Setup for Case Studies

To verify DREAM performance in prediction of Icelandic dust transport for different transport scales, we setup the model domain to cover sufficiently large area surrounding Iceland, as shown in Figure 6.

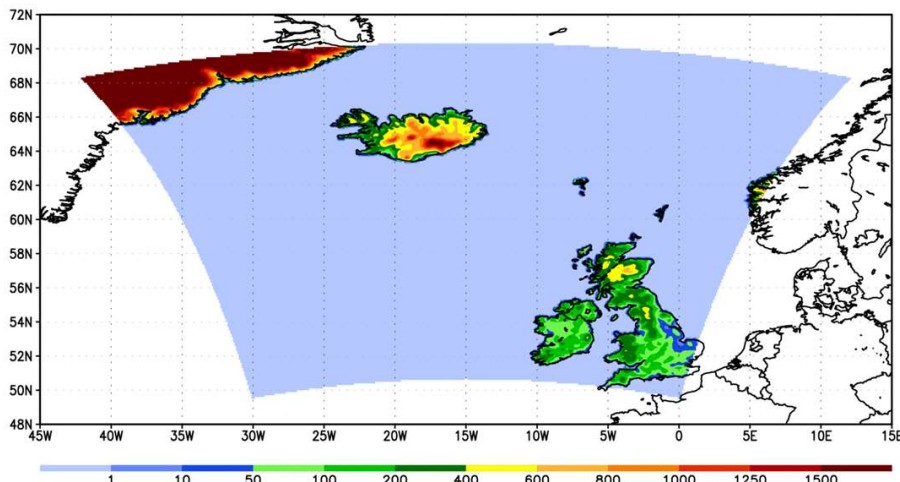

**Figure 6.** Domain represented as topography height in meters above sea level.

The model has 28 model vertical levels spanning from the surface to 50 hPa. The longitude–latitude coordinates are rotated in the model in such a way that the coordinate origin is located in the middle of the model domain. In this way, the reduction of the longitudinal grid-size is minimized as the southern and the northern boundaries of the domain are approached, and, therefore, longer model time steps are used to increase the processing time efficiency. In the transformed coordinate system, the model horizontal resolution is set to 0.05 deg, in which model grid point distance is approximately 3.5 km within the domain. The non-hydrostatic character of model dynamics [63] provides the vertical atmospheric dynamics to include convective dynamics as well. The model's basic time step (forecast time increment) is 18 s. Dust advection and lateral diffusion are computed every second time step, while physical processes, including dust surface emission and vertical diffusion, are updated every 4th time step. The large-scale precipitation is calculated every 8th time step.

DREAM forecast was done for three selected dust transport episodes: in 2011, 2018 and 2019. Forecast period for each case study was four days. The airborne dust content at initial time was set to zero, and it is referred as the "cold start" condition. This is done because there are no satisfactory three-dimensional dust observations to be assimilated for initialization of dust forecast. For subsequent days 24-h concentration forecast from the previous day is used as the initial state. The initial and boundary conditions for the atmospheric model part are specified using the numerical weather prediction parameters of the ECMWF global model (European Centre for Medium-Range Weather Forecasts).

## 3. Results

DREAM model performance is validated and evaluated against available observations for three severe dust events. The 2011 dust transport episode mainly affected Iceland and its neighborhood, while the other two cases in 2018 and 2019 correspond to long–range dust transport towards Ireland, and the British and Faroe Islands.

The choice of case studies is governed by the available observations and characteristics of the events. The main constraints of dust events information in high latitudes are: scarce in situ measurements, high cloud coverage, which impacts satellite dust information, and lack of evidence derived from the general public because of low population. The available observations on airborne dust during selected dust storm cases are used for model performance validation as follows (Appendices B and C):

- Horizontal distribution of dust-related parameters evaluated by the model and the MODIS (Moderate Resolution Imaging Spectroradiometer) observations;

- Model cross-section and point-based vertical profiles vs. aerosol extinction coefficient profiles retrieved from the CALIPSO (The Cloud–Aerosol Lidar and Infrared Pathfinder Satellite Observation);
- Model time–height cross-sections of dust concentration for selected locations vs. in situ ceilometers attenuated backscattering profiles;
- Model near-surface concentration vs. in situ particulate matter ($PM_{10}$) data;
- Air mass back trajectories used as auxiliary information on the analysis of long-range dust transport.

*3.1. September 2011 Case—Short–Range Dust Transport Episode*

The September 2011 case is selected to describe dust hot-spot activation and intense short-range dust transport. The event started on 9 September 2011 by activation of multiple dust sources, caused by relatively intense N–NE winds over the whole island. According to NASA MODIS satellite observations (Terra/Aqua corrected reflectance (true colour) and Aerosol Optical Depth images, available at https://worldview.earthdata.nasa.gov/, accessed on 25 May 2022), the sequence lasted for almost 5 days, but the most intense event which affected Reykjavik occurred on 12 September. Although there were multiple dust plumes, the one originating from Hagavatn area mostly affected the Iceland capital. This 10 km$^2$ wide hot-spot is located about 70 km NE from the city. Additional material might have been resuspended relicts of volcanic ash from the Grimsvötn eruption on 21–28 May 2011. The plume mainly affected the S–SE areas of the island and the maritime areas off the southern coasts. Severe dust storm had been detected in Reykjavik during the afternoon hours on 12 September 2011. The dust plume was captured by the NASA satellites Terra and Aqua, which is clearly seen on the MODIS true color (Figure 7a) and AOD images (Figure 7b).

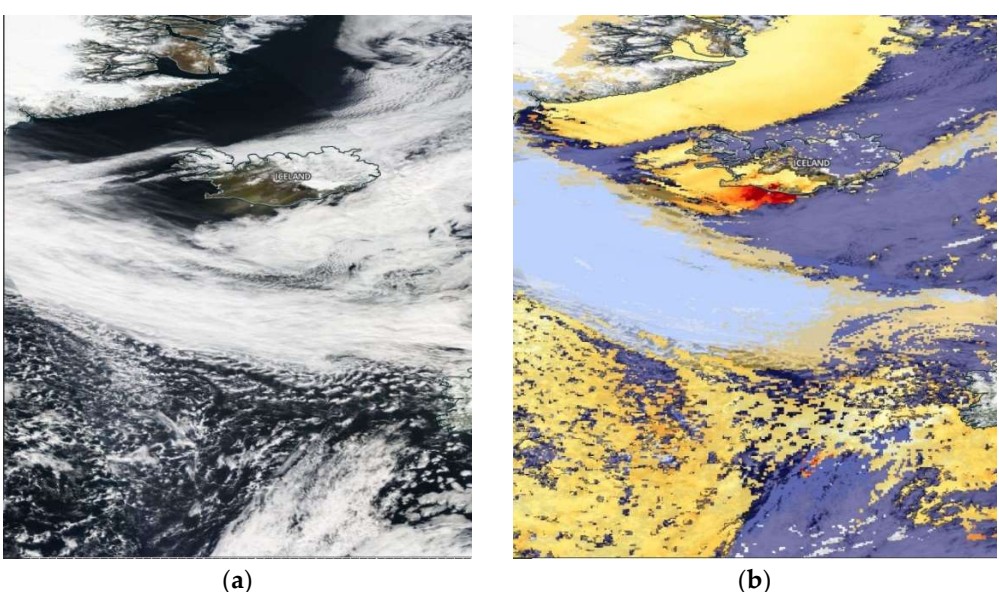

(**a**)                                                           (**b**)

**Figure 7.** Dust storm event in Iceland on 12 September 2011. Dust plume detected: (**a**) MODIS Terra and Aqua combined corrected reflectance (true color); and (**b**) Aerosol Optical Depth (AOD).

A large amount of dust, emitted from a nearby Hagavatan hot-spot and transported by intense N–NE winds, caused an increase in $PM_{10}$ concentration at Reykjavik (Grensásvegur) station in the afternoon hours of 12 September 2011 (Figure 8). We anticipate that dust concentration contributed to the total measured particulate matter, because there are usually no other PM sources activated in this period of the year.

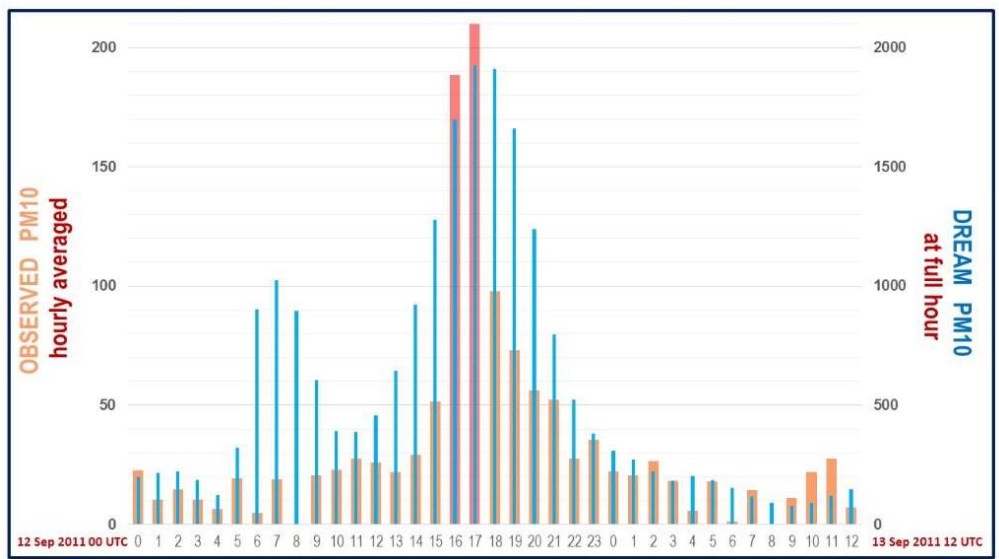

**Figure 8.** Reykjavik dust storm evidence, time period 12 September 2011 00UTC–13 September 2011 12UTC; orange and red bars: Hourly averaged observed $PM_{10}$ concentration ($\mu g\ m^{-3}$); blue bars: predicted surface $PM_{10}$ dust concentration ($\mu g\ m^{-3}$).

The main findings are: the peak of predicted dust concentration at 17UTC on 12 September 2011 coincides with the observed $PM_{10}$; the timing of the dust storm maximum gust was captured; the model has the ability to predict sudden increases and decreases in dust concentration. The secondary maximum predicted in the morning hours was not observed. Model data are presented as instantaneous full-hour values, while observations are hourly averages. Uncertainties of model results, including the false signal for dust $PM_{10}$ in the morning hours, can be mainly caused by somewhat shifted wind patterns with a consequent shift of dust plumes, which is related to weather forecast quality, and by uncertainty in the source spatial patterns and soil texture information. Due to the high spatial and temporal variability of dust concentrations during dust storms using high resolution models, point-on-point verification may cause a double penalty problem, when model scores decrease with resolution increase [47]. Average PM measurements in severe short lived dust storms can lessen the maximum dust concentration gusts by the order of magnitude, when is expected model data (outputs at specific moment) to overestimate observed means. In this case, model overestimated observed averages, but for the whole day. Since the model captured the timing of maximum dust gusts, some of the mentioned uncertainties must be responsible for the overestimation. In the future, when more similar cases will become available, field studies and/or observations come into play and a clearer assessment of the uncertainty source will be provided.

Model dust concentration for the event for three successive days (11–13 September) is shown in Figure 9. Evolution of three dust plumes generated by strong NE surface winds is clearly visible over the period: their formation, extreme stages and finally their decay. Dust concentration on 12 September reached 2000 $\mu g\ m^{-3}$ in Reykjavik, and even 3000 $\mu g\ m^{-3}$ in the vicinity of the city. Similar values have been measured in Iceland during severe dust storms [1,22,30].

The ability of the model to clear up the atmosphere of high dust concentrations after such dust storms is also an important feature. Twenty-four hours after the achieved maximum dust concentration, the dust plume mostly disappeared, and the remaining concentrations in the area caused dust haze during 13 September.

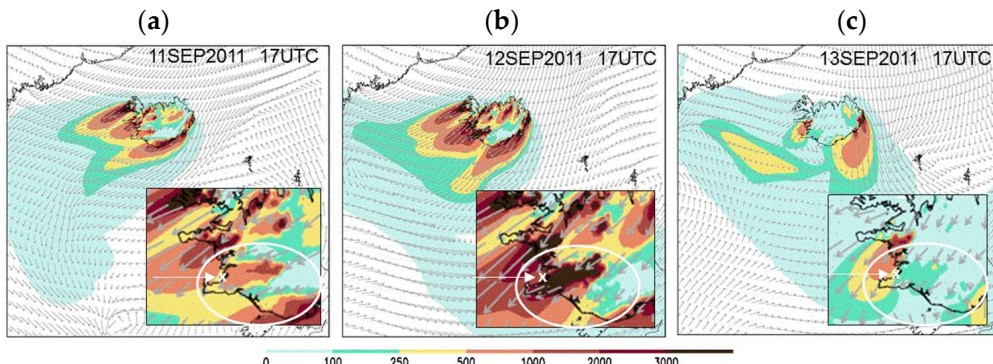

**Figure 9.** DREAM surface dust concentration (μg m$^{-3}$) forecast for September 2011 dust storm: (**a**) The day before the dust storm peak (11 September at 17UTC); (**b**) at the dust storm peaking hour (12 September 17UTC); and (**c**) the day after dust storm peak (13 September 2011 17UTC). Reykjavik location is marked with "x" and pointed to with a white arrow. Region within the circle is a dust plume zone.

While crossing the western Iceland coastline region on 12 September 2011, CALIPSO recorded a dust pattern originating from the island. Figure 10 depicts the MODIS–Aqua AOD at 550 nm (colored map), the CALIPSO overpass (red line on Figure 10b), while the red thick line corresponds to the part of the CALIOP track that has been considered for the curtain plots shown in Figures 10 and 11. Along the same CALIPSO path, we present the predicted dust concentration in the lower atmosphere (Figure 11a). The model dust pattern well compares with the observed profile of the CALIPSO classification of atmospheric features, which reports aerosols extending within the first 3 km of the atmosphere (Figure 11b). The top of the dust layer was gradually decreasing northward, as evident from both observed and modeled vertical profiles.

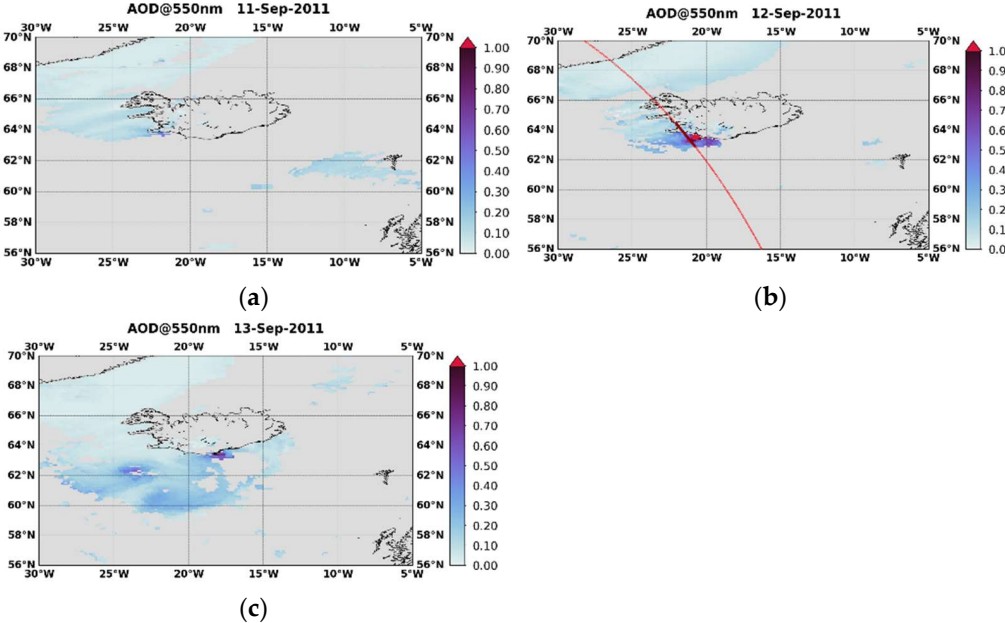

**Figure 10.** (**a**–**c**) MODIS–Aqua AOD at 550 nm representing dust pattern over the period of 11–13 September 2011. The red line on the 12 September (**b**) image shows the only available CALIPSO overpass.

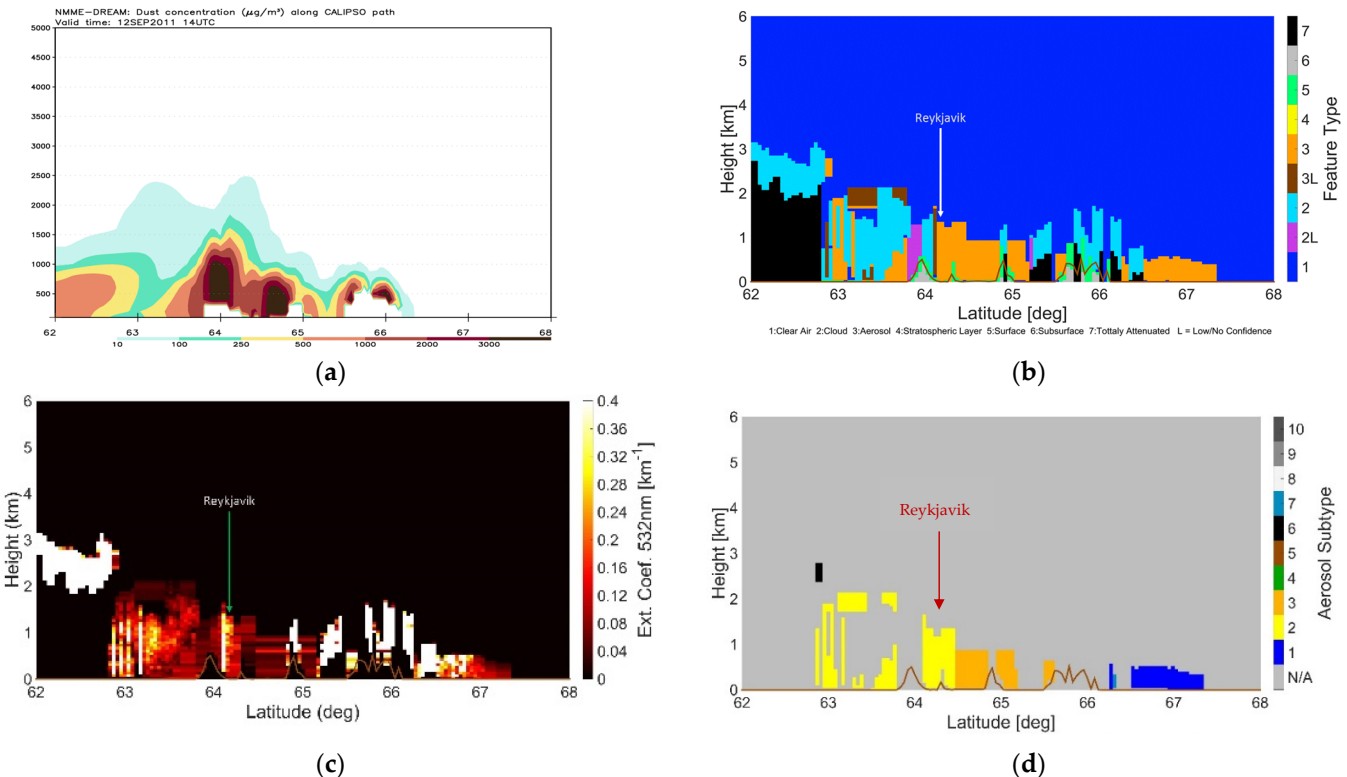

**Figure 11.** DREAM model and CALIPSO data along satellite path closest to the event, for 12 September 2011: (**a**) Model dust concentration; (**b**) CALIPSO vertical profile of recorded feature type with value 3 (aerosol) dominant at detected dust plume region; (**c**) CALIPSO extinction coefficient that marks detection of the pure dust near Reykjavik; and (**d**) CALIPSO aerosol subtypes: 1—marine, 2—dust, 3—polluted continental/smoke, 4—clean continental, 5—polluted dust, 6—elevated smoke, 7—dusty marine, 8—PSC aerosol, 9—volcanic ash, 10—sulfate/other.

Note that all three major areas/plumes (at 64° N, 65° N and 66° N), detected by CALIPSO algorithms as a dust aerosol, are predicted by the model with almost no temporal and location bias. The mean height of the dust layer, according to CALIPSO measurements, is about 1 km, with small area between 62° N–63° N reaching 3 km altitude; DREAM reproduces the same sequence (Figure 11a,b). The dust/aerosol pattern near 67° N is out of our focus since it originates from sources outside the Iceland. Similar observations of Icelandic dust plume height were detected by CALIPSO and Light Aerosol Optical Counter in vertical profiles in 2016 [67].

The Icelandic dust event on the 12 September 2011, was captured by CALIPSO, as shown in Figure 11. CALIOP observations and the DREAM model forecast are in good agreement, both quantitatively and regarding the horizontal and vertical extend of the aerosol layer. The tropospheric features classified as aerosols are further sub-classified as dust or polluted continental subtypes, mainly due to the relatively high particulate depolarization ratio and the low elevation of the detected layer [68]. It is important though to clarify that a significant part of the observed heavy dust event, as captured by CALIPSO, is possibly misclassified as cloud on Figure 11b, where the cloud–aerosol discrimination (CAD) score (not shown) reveals a relatively low level of confidence in the feature type classification. The misclassification of the observed dense dust event possibly relates to the set of the multidimensional probability density functions (PDF) [69] implemented under the CALIOP operational cloud–aerosol discrimination algorithm (COCA) and the scene classifier algorithm (SCA) [70], towards differentiating between clouds and aerosols [71,72]. The special characteristics of dust storms, such the high values of the attenuated backscatter and volume depolarization ratio at 532 nm, in conjunction with the relatively low temper-

ature and high-latitude of the event where infrequent dust events are observed [73–75], compromise the full identification of high-latitude dust. However, the improvements made in the framework of CALIPSO Version 4 algorithms [69,76] still result mostly in dust aerosol subtype classification (Figure 11d).

Profiles of observed and predicted dust aerosol extinction coefficient at 532 nm, both averaged over the CALIPSO path are shown in Figure 12. The DREAM model (black line) successfully simulates the CALIOP-based pure-dust extinction coefficient averaged profile (red line), reproducing the vertical structure both qualitative and quantitative, falling well within the variability of the transported dust plume ($\pm 1$ $\sigma$—shaded area). Such low dust layer is typical for high latitudes, as a consequence of the dominating shallow mixing layer conditions [77].

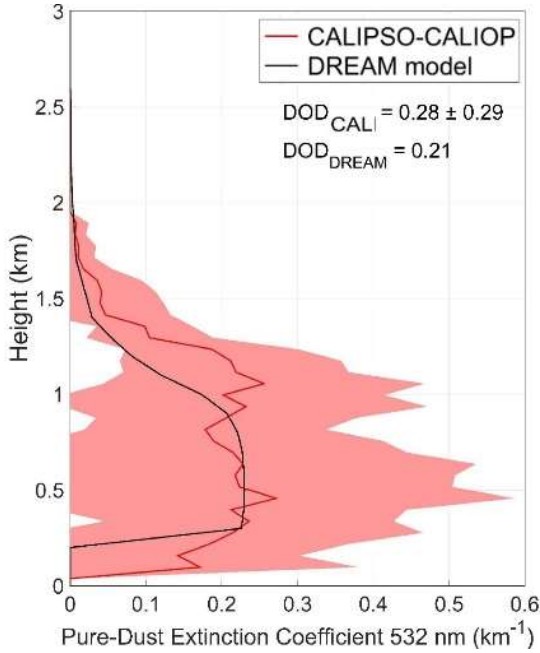

**Figure 12.** CALIPSO (CALIOP) (red line) and DREAM (black line) pure dust extinction coefficient at 532 nm vertical profiles averaged over the CALIPSO path for 12 September 2011.

### 3.2. September 2018 Case—Long–Range Dust Transport Episode

This dust event was captured by MODIS on 20 September 2018, showing at least 3 narrow dust plumes on the south coast of Iceland, visible both on true color and AOD images (Figure 13). Northern winds with velocities higher than 10 ms$^{-1}$ over south Iceland lifted dust from the Landeyjarsandur and Markarfljót dust sources, and from the Myrdalssandur, Meðallandssandur and Jokulhlaup sediments from Skafftá river area. Skeidararsandur deposits were likely activated too.

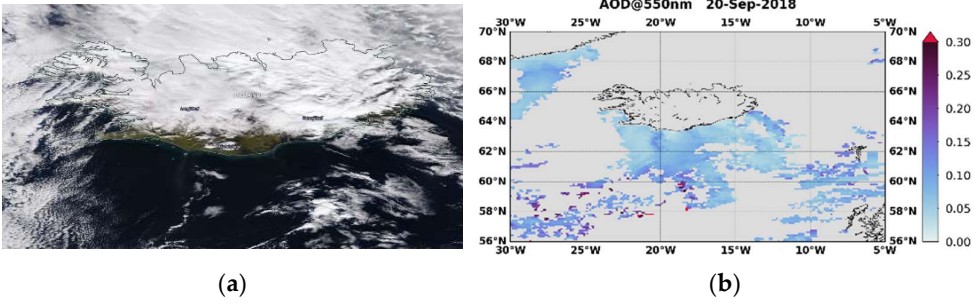

(**a**)　　　　　　　　　　　　　　　　　　　　　(**b**)

**Figure 13.** The evidence of the dust storm event in Iceland on 20 September 2018. Detected dust plumes reached Faroe Islands next day: (**a**) MODIS true color; and (**b**) MODIS AOD.

The model predicted massive dust storm on 20 September 2018, lasting more than 24 h (Figure 14a,b) and the maximum surface dust concentrations reached Faroe Islands on 21 September (Figure 14b). Another branch of the same system arrived at the British Islands on the afternoon hours of the same day. Atmospheric circulation characterized by intense north–west winds, dispersed dust towards the Faroe and British Islands as indicated also by the MODIS AOD images (Figure 14c,d).

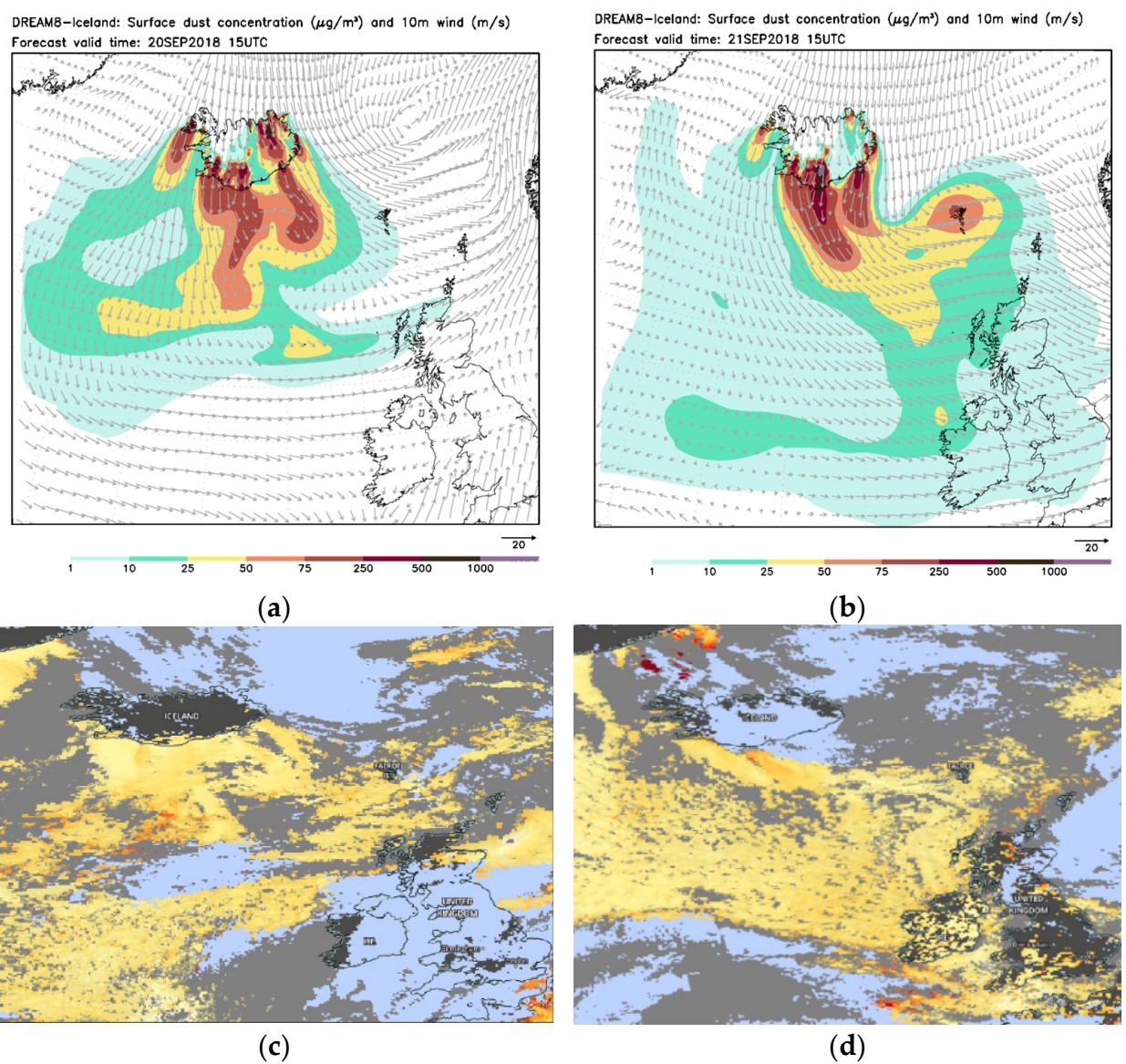

**Figure 14.** Dust storm 20–21 September 2018. (**a–c**) DREAM surface dust concentration; and (**c–d**) MODIS AOD.

The peak and timing of simulated surface dust concentration on 21 September over the Faroe Islands well agree with the observations at Torshavn PM Station (Figure 15a). Time–height evolution of the simulated dust passage over Torshavn shows that the dust layer is mainly confined below 1.5 km, as displayed on Figure 15b. The model uncertainties in this case, i.e., the difference between observed and modeled concentrations, are also the consequence of the different factors mainly related to the wind filed simulation and source specification in the model as explained in the analysis for the case study of the short range transport. In this case, with long range transport verification, significant dust is transported also over higher altitudes. The model also must distribute the dust properly in the vertical direction as in the horizontal.

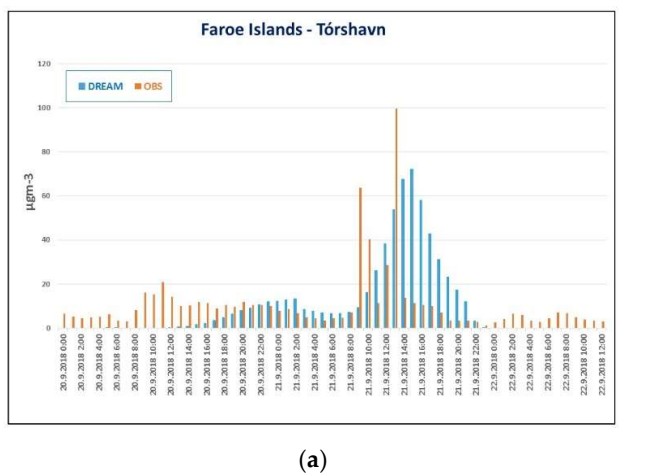

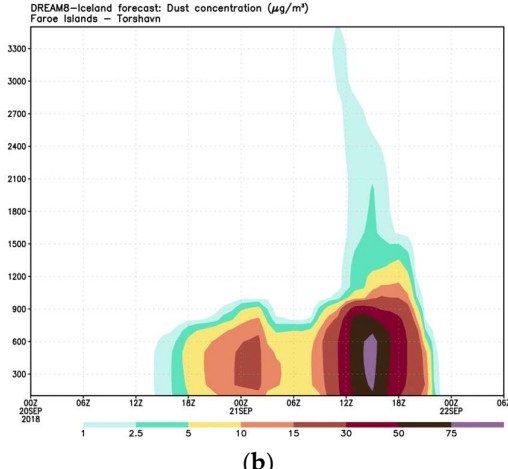

(**a**)

(**b**)

**Figure 15.** Faroe Islands (Torshavn)—Dust episode 21–22 September 2018: (**a**) Observed (orange) and predicted (blue) PM$_{10}$ concentrations (μg m$^{-3}$); and (**b**) time–height plot of dust concentration (μg m$^{-3}$).

For the two selected stations in Great Britain (Chilbolton) and Ireland (Mace Head), we present the height–time evolution of the dust event as represented by ceilometer observations (Appendix D) and model data (Figures 16 and 17, respectively).

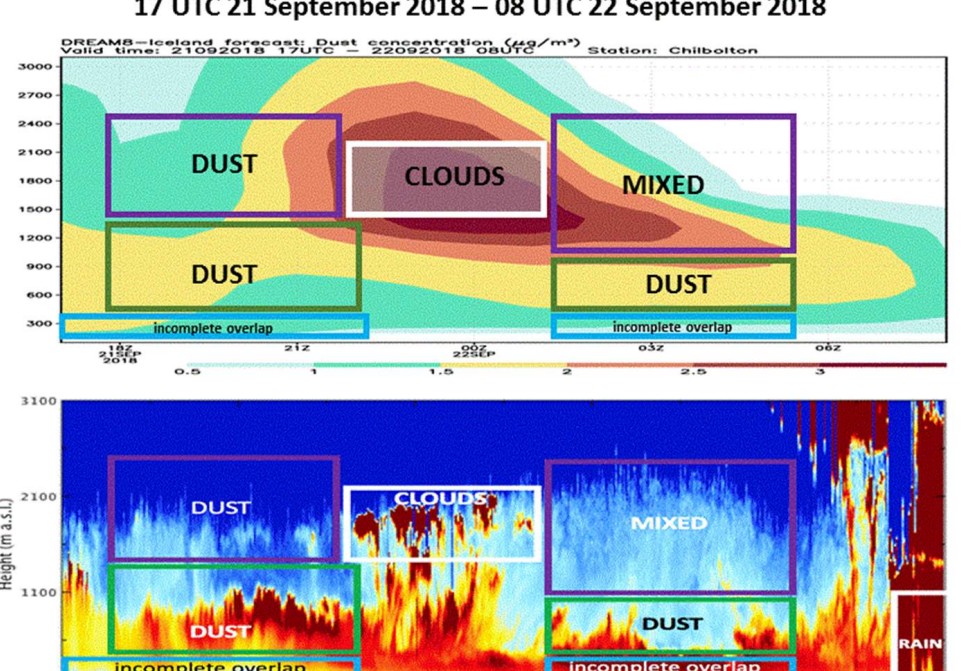

(**a**)

**Figure 16.** *Cont.*

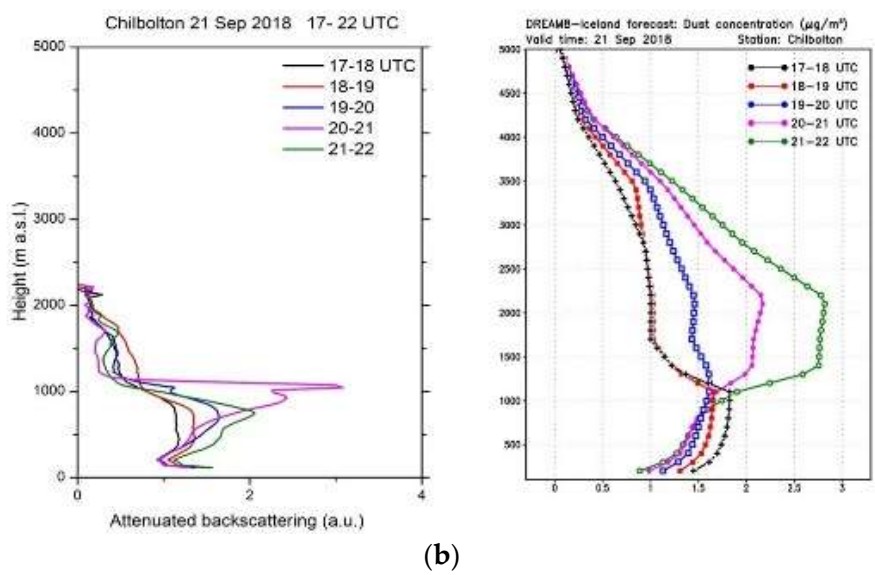

**(b)**

**Figure 16.** Chilbolton station: (**a**) Time series of the model dust concentration (μg m⁻³) (upper) and ceilometer attenuated backscatter (lower), along with aerosol type classification inferred using air mass back trajectories; and (**b**) vertical profiles of the attenuated backscattering calculated from the ceilometer observations (left) and of the dust concentration (μg m⁻³) predicted by the model (right) for selected time intervals. Note: Boxes in the model images are approximately positioned in accordance to those specified in the LIDAR/ceilometer analysis plot.

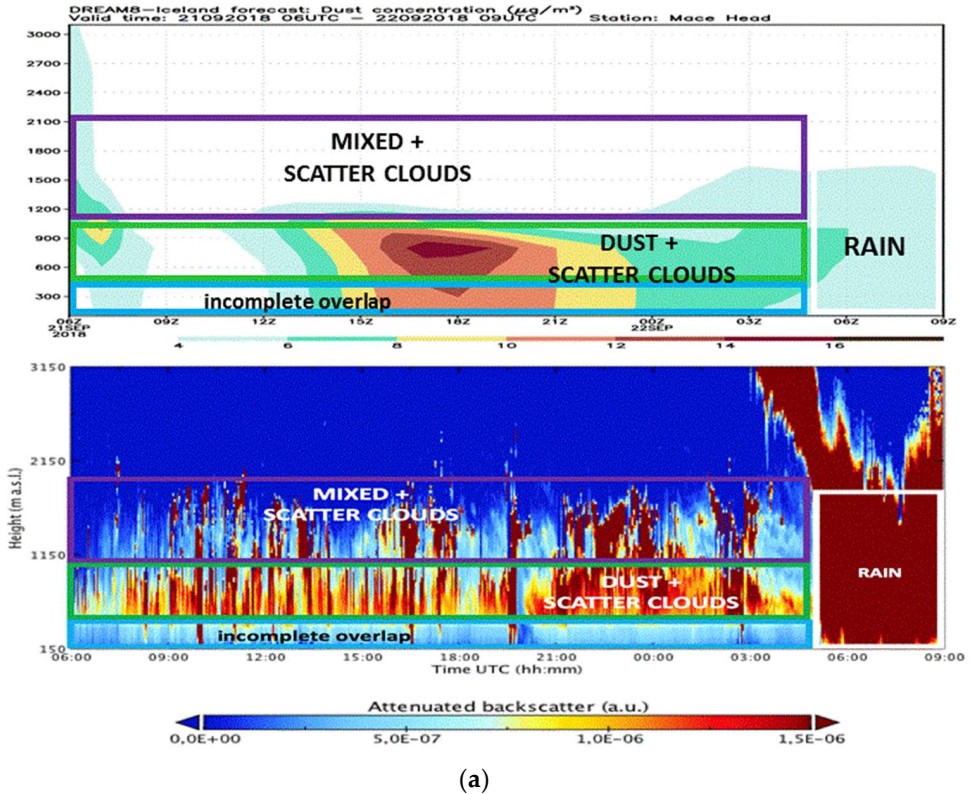

**(a)**

**Figure 17.** *Cont.*

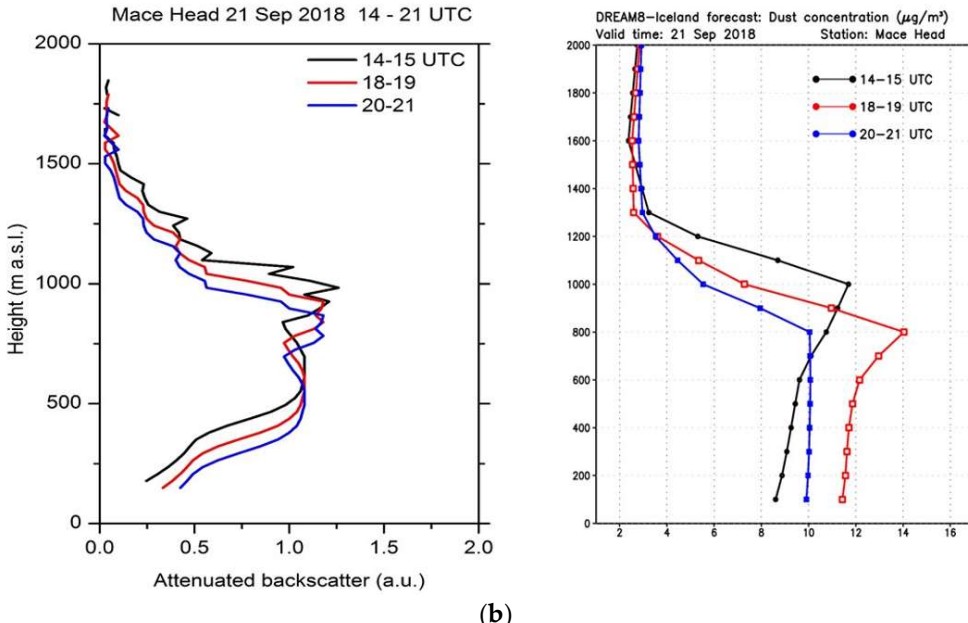

**Figure 17.** Mace Head station: (**a**) Time series of the model dust concentration (µg m$^{-3}$) (upper) and ceilometer attenuated backscatter (lower), along with aerosol type classification inferred using air mass back trajectories; and (**b**) vertical profiles of the attenuated backscattering calculated from the ceilometer observations (left) and of the dust concentration (µg m$^{-3}$) predicted by the model (right) for selected time intervals. Note: Boxes in the model images are approximately positioned in accordance to those specified in the LIDAR/ceilometer analysis plot.

Ceilometer measurements from the Chilbolton (51.15° N, 1.44° W) ACTRIS Cloudnet station (www.actris.eu, accessed on 25 May 2022), for the 20–22 September 2018 case, were used to validate the modelled dust concentration (Figure 16). The ceilometer is a Vaisala CL51 operating at 910 nm [78] and calibrated following the procedure described in [79], which allows providing the time series of the vertical profile of the aerosol attenuated backscatter.

The left panel (Figure 16a) shows modeled (upper) and ceilometer (lower) height–time evolution of the event over the time window from 17 UTC on 21 September to 08 UTC on 22 September 2018. The boxes overlapped with the ceilometer time series shows the atmospheric target classification along with the aerosol type: three main aerosol layers are observed peaked at around 900 m, 1100 m and 1700 m above sea level. The aerosol classification was carried out using the air mass back trajectories (not shown) from HYSPLIT NOAA model [80]. Each layer has been classified as dust or mixed aerosol, depending on the height of corresponding trajectory passing over Iceland. As shown in Figure 16a, the model dust concentration fits well with the observed time–spatial patterns. The majority of the predicted dust mass (upper plot on Figure 16a), associated with the developed cloud system, reached Chilbolton around 22 UTC on 21 September 2018. According to ceilometer data, dust layers tend to remain lower in height until the rain occurrence around 08:00 UTC on 22 September 2018. In addition, we compare the vertical profiles of the observed attenuated backscatter coefficient with the modelled dust concentration averaged over five consecutive hourly intervals between 17 and 22 UTC (Figure 16b).

The comparison, although semi-quantitative using the attenuated backscatter coefficient as a proxy of the aerosol concentration in the atmosphere, shows that:

- The aerosol dynamics provided by the model is in good agreement with the observations with a difference within ±1 h in the detection of the main features of the dust event such as beginning, peak concentration, dynamical evolution;
- The model also detects a dust concentration at altitudes above 2.0–2.5 km above sea level, especially on the 21 September 2018, which is not completely consistent with the

observations and may be due to either a model overestimation or to the ceilometer's sensitivity at small dust concentration, which is more common than in case of lidar observations [81].

No PM observations in Iceland and appropriate CALIPSO crossings are available for this dust event. In general, the model reproduces the major features of the dust episode observed at measurement sites distant more than 1000 km from the source region.

For the 20–22 September 2018 case, ceilometer measurements from the Mace Head (51.15° N, 1.44° W) ACTRIS Cloudnet station (www.actris.eu, accessed on 25 May 2022) have been used to validate the modeled dust concentration (Figure 17). Mace Head ceilometer is a Lufft CHM15k operating at 1064 nm [81]. This is one of the most powerful models available on the market, mainly due to its laser source. Time series of the vertical profile of the aerosol attenuated backscatter are obtained applying a calibration factor.

The left panel (Figure 17a) shows modeled (upper) and ceilometer (lower) height–time evolution of the event over the time window from 06 UTC on 21 September to 09 UTC on 22 September. The boxes overlapped with the ceilometer time series shows the atmospheric target classification along with the aerosol type: two main aerosol layers are observed peaking at around 500 m and 900 m above sea level. The classification was carried out using the air mass back trajectories (not shown) from HYSPLIT NOAA model [80]. Cloud and rain can be clearly identified by high values of the attenuated backscatter coefficient in the boundary layer and close to surface, respectively, and by the extinction of the signal above. Aerosol type is classified as dust or mixed aerosol depending on the height of corresponding trajectory passing over Iceland. As shown in the figure, dust layers tend to increase in altitude and concentration from 15 UTC until the rain occurrence on the morning of 22 September. The model dust concentration is in very good agreement with the observed time–spatial patterns both for the altitude of the layer and its concentration. The comparison of vertical profiles of the observed attenuated backscatter with the model dust concentration between 14 and 22 UTC is also shown on Figure 17b. The frequent occurrence of clouds does not allow to get a sufficient statistic in other hourly intervals during the same time window.

Assuming the attenuated backscatter coefficient as a semi–quantitative proxy of the aerosol concentration in the atmosphere, the comparison shows that:

- The aerosol dynamics provided by the model is in good agreement with the observations for the detection of the largest part of the aerosol event, in the timing of the peak concentration and in the maximum altitude reached by the dust. The most relevant difference is at the end of the event when from the observations keeps a strong signal until 03 UTC on 22 September, while the model shows a smaller concentration already at 00 UTC on the same day;
- The profile comparison shows an almost perfect match between the shape of the attenuated backscattering profiles and the model concentration. This is true both for the altitudes of the layers and for the relative difference in their signals (i.e., attenuated backscatter for the ceilometer and dust concentration for the model).

This long range transport sequence shown by the NOAA HYSPLIT trajectory (Figure 18a) indicates the arrival of the Icelandic dust to the British Isles in the late afternoon hours of 21 September 2018. The dust reached Mace Head (Ireland) after traveling more than a thousand kilometers over the ocean. The comparison of the observed and modelled concentrations is shown on Figure 18b. The increased model values, during several afternoon and evening hours on 21 September 2018, agree with the observed pattern. Then, the concentration decay during early morning hours on 22 September is captured by both profiles, along with the secondary maximum in the afternoon hours of the same day.

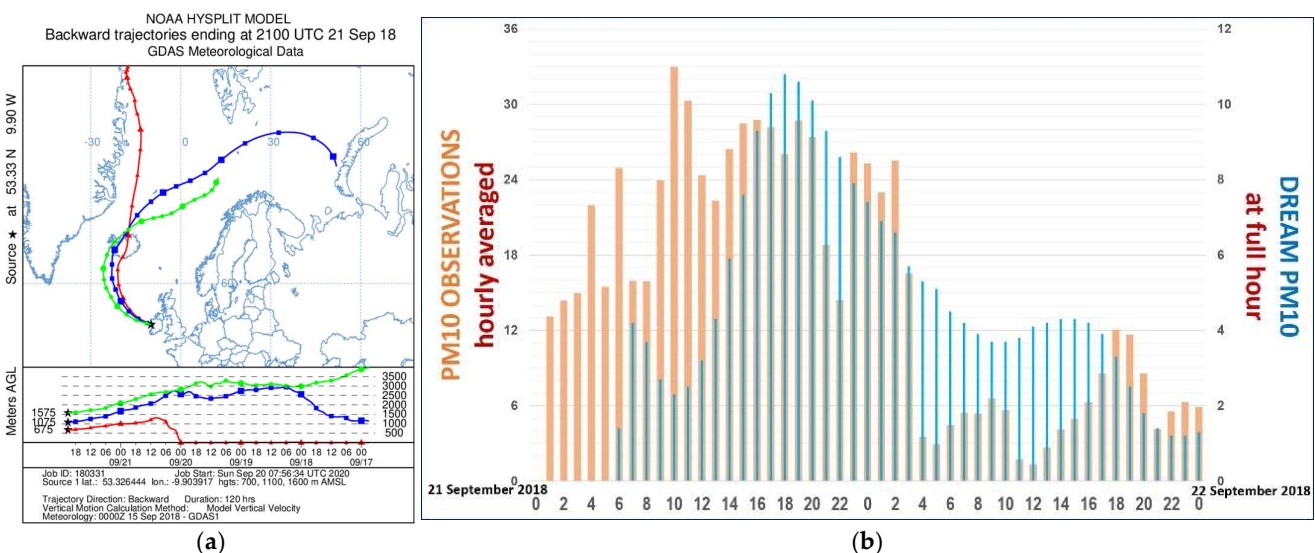

| (a) | (b) |

**Figure 18.** Mace Head station; (**a**) NOAA HYSPLIT model trajectory evidence on Icelandic dust arriving at Mace Head at 21 UTC 21 September 2018; and (**b**) observed PM$_{10}$ (orange) and model surface concentration (blue) (µg m$^{-3}$) over the period 21–22 September 2018. The red, blue and green lines represent simulated trajectories from the HYSPLIT model. Each line represent air parcels released from 3 different heights above ground level. Parcels are transported by the mean 3-D wind field, provided by the atmospheric model.

*3.3. October 2019 Case—Long–Range Dust Transport Episode*

This dust event was captured by MODIS, showing visible dust plume extending to about 400 km towards the British Isles (Figure 19a,b). Multiple dust hot-spots were activated on 24 October 2019 with strong north winds with velocities higher than 20 ms$^{-1}$ over the southern Iceland (Myrdalssandur, Landeyjarsandur, Skeidararsandur, and Jokulhlaup sediments from the Skaftá river area). Dust remained in the atmosphere between Iceland and the British Isles until 26 October 2019.

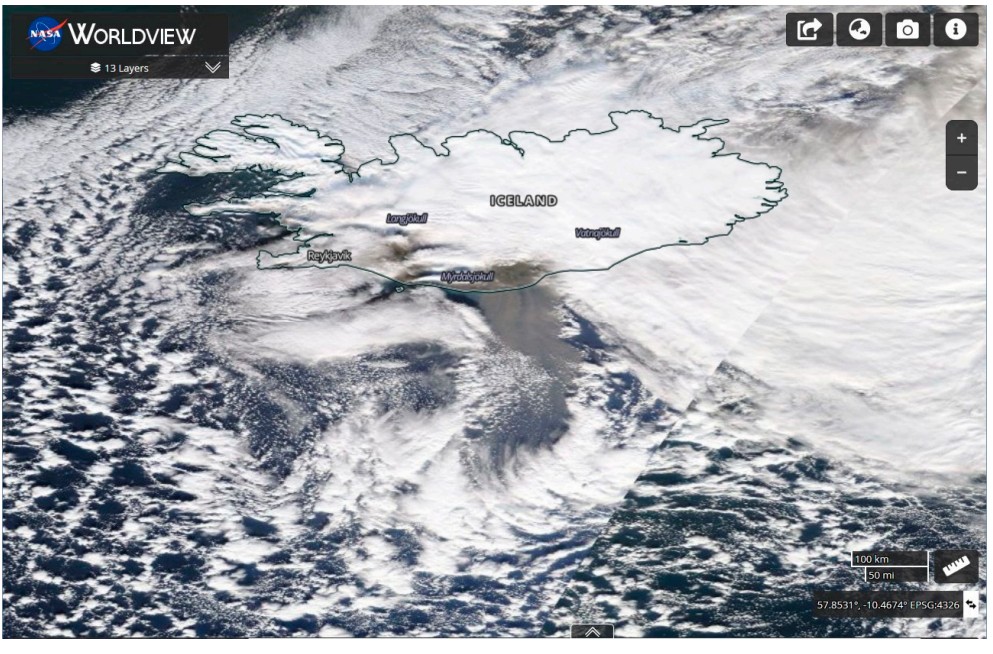

(**a**)

**Figure 19.** *Cont.*

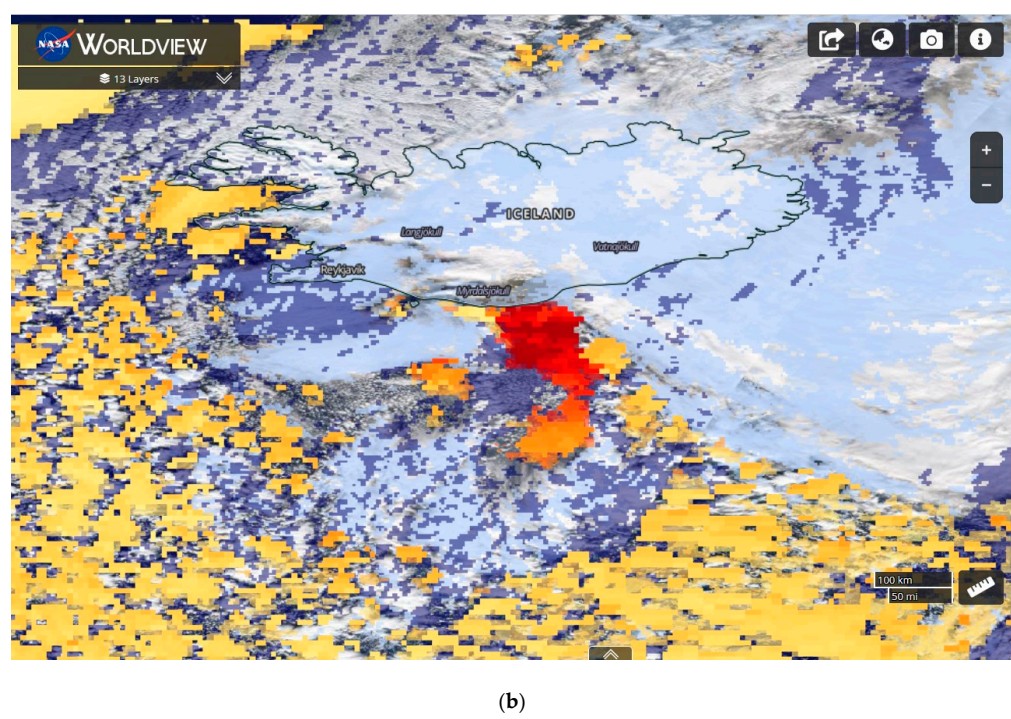

(**b**)

**Figure 19.** MODIS satellite observed dust plume on 24 October 2019. (**a**) True color; and (**b**) AOD.

The model predicted the massive dust storm reaching North Ireland, together with dust driven by a secondary cyclonic circulation orbiting around the Faroe Islands on October 25 (Figure 20a–c). These dust patterns are consistent with the structure captured by NASA MODIS AOD images (Figure 20d–f).

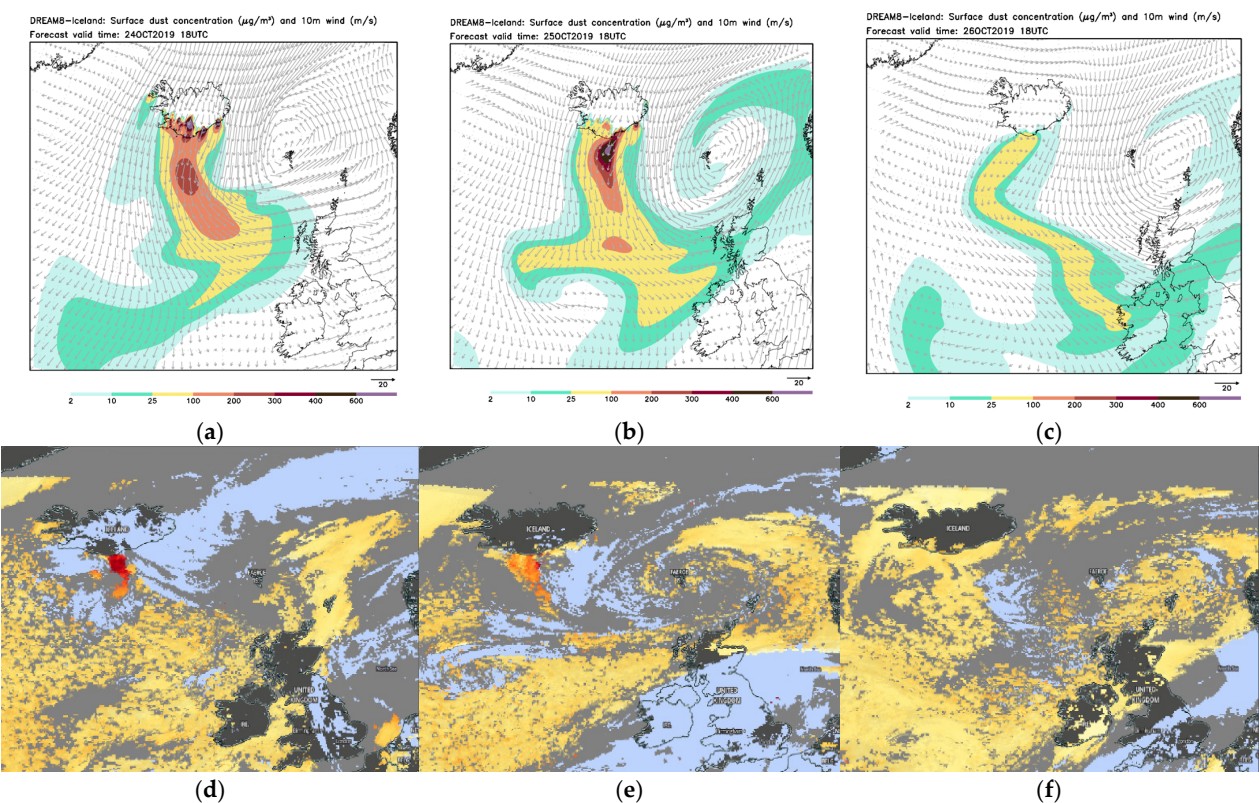

**Figure 20.** Dust storm 24–26 October 2019. (**a**–**c**) Model surface dust concentration (μg m$^{-3}$); and (**d**–**f**) MODIS AOD.

According to the HYSPLIT trajectories (Figure 21a,b), which are in compliance with the predicted surface dust concentration and the observed PM10 at Mace Head station (Figure 21c), the Icelandic dust approached Ireland in the evening hours of 25 October 2019 and it persisted all the next day. The predicted dust concentration shows maximum values for 26 October, as confirmed by the observed PM10, although the first peak in the model profile (early morning hours on 26 October) is overestimating the observed PM values. The model also produced an increased concentration before the observed increase, most probably because of the same reasons, as in previous case studies (wind-filled forecast, sources specification and point-on-point verification approach).

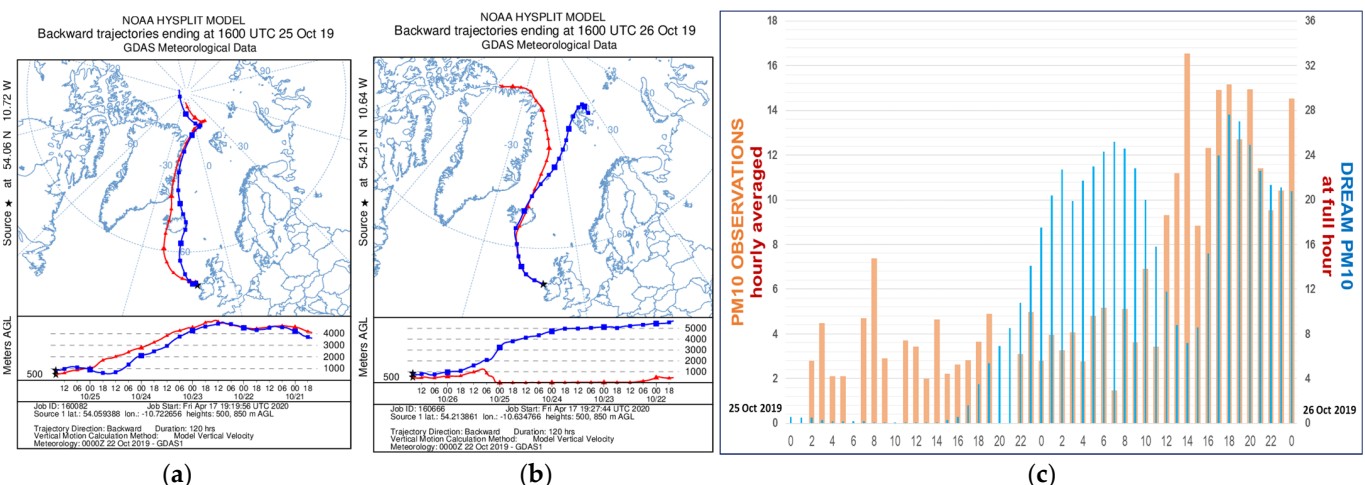

(a)  (b)  (c)

**Figure 21.** The same explanation as for the Figure 18. Here we show the trajectories for 2 different release heights. The evidence of dust arriving Mace Head station (Ireland) 25–26 October 2019; (**a**) and (**b**) NOAA HYSPLIT backward trajectories for 25 and 26 October 2019, respectively; and (**c**) observed PM$_{10}$ (orange bars) and model surface PM$_{10}$ dust concentrations (blue bars) (μg m$^{-3}$).

## 4. Conclusions

Being constantly exposed to intense glacial and aeolian activity, Icelandic soils and volcanic sediments constitute the largest European dust source, and one of the most productive sources of mineral dust at high latitudes. Current operational dust transport models, with typical horizontal resolution of about 10 km, cannot recognize such sources and their physical characteristics, specifically small-scale highly emissive areas defined as "hot-spots". By combining information on Icelandic topsoil characteristics from numerous field campaigns, we developed a high resolution dust source mask with specific particle size distribution and emissivity potential for each source area. It represents the main input parameter for the upgraded fully dynamic coupled atmosphere–dust emission and transport model (DREAM). In this research, three case studies were selected to assess the model quality performance in simulating the Icelandic dust transport over various time–spatial scales.

Selected case studies of dust transport demonstrated the model's capability to forecast both large- and small-scale dust transport episodes. All major features, such as timing, horizontal and vertical distribution of the processes, as well as capability to maintain such patterns over long distances, are in accordance with the observations.

The dust episode in September 2011, which mainly affected Iceland and its neighboring area, represented a case of intensive short range dust transport. Analyzing satellite products such as MODIS (Terra/Aqua), Corrected Reflectance (True Color) and Aerosol Optical Depth, we showed that the model is capable to reproduce complex horizontal dust patterns as well as timing of the process. The model dust extinction coefficient at 532 nm and dust concentration vertical profiles demonstrated good agreement with corresponding CALISPO profiles of extinction coefficient and aerosol/feature types. Comparison between observed and modelled surface PM$_{10}$ concentration for Iceland's capital Reykjavik proved

the model's ability to predict relatively rapid increases and decreases of particles concentration, which represents one of the main characteristics of the short–lived dust storms.

Dust episodes in September 2018 and October 2019, with Icelandic dust aerosol reaching Great Britain, Ireland and the Faroe Islands, have been chosen to evaluate model performance in forecasting long-range dust transport that originated from small-scale hot-spot sources. Model surface dust concentration patterns were confirmed by the NASA MODIS satellite products such as (Terra/Aqua) Corrected Reflectance (True Color) and Aerosol Optical Depth. Ceilometer attenuated backscatter measurements at Mace Head (Ireland) and Chilbolton (UK), and related aerosol type classification, are in good agreement with corresponding structure of the dust layer provided in the model dust concentration profiles. Atmospheric dust transport over the Atlantic, characterized by the fact that the majority of the dust mass is concentrated in the lower atmosphere (below 3 km), is also successfully simulated with DREAM model and confirmed by ceilometer/Lidar observations. The comparison between the observed and modeled surface $PM_{10}$ concentrations for the abovementioned stations showed that the prognostic system is able to forecast the timing of dust reaching areas after traveling even thousands of kilometers from sources with a time accuracy of about 2 h. NOAA HYSPLIT backward trajectories are also in accordance with the modeled results of these long-range Icelandic dust transport events.

This study confirmed the findings of emerging studies on long-range transport of Icelandic dust (and HLD generally) towards the High Arctic and Europe [16,20,21,28] and Icelandic dust impacts on climate [42,82]. The evidence of long-range transport during high latitude dust storms is scarce, but such events are noted as an important climate system feature in recent literature. Better understanding of these storms can contribute to the future development of high latitude dust forecasting and implementation of aerosol component in climate models, because of its potentially important connection to changing climate.

Analysis of the selected case studies also highlights the limitations of the dust model to predict the observed concentrations at the location with observations. Due to the scarce surface $PM_{10}$ observations, and no available wind measurements at the same location or in the vicinity and over source area, it remains unknown if the cause for the false dust peaks is the local overestimation of the surface wind velocity, wind direction or specification of the dust sources and its characteristics in the model. The approach in the verification of dust model using surface measurements, here called point-on-point verification, in cases when there is a large spatial variability of dust concentrations, could be misleading. The methodology for such verifications should be revised to avoid misleading conclusions about model bias when dust plumes are somewhat displaced compared to their propagation in the reality. Thus far, for the dust forecasting in general and especially for the forecasting of local severe dust storms, the best expected dust forecast quality remains unknown.

This study demonstrated the ability of the model to predict features of both long-range and short-term transport of high latitude dust, thus opening an opportunity for accessing various interactions with the environment and climate. These are dust effects to marine geochemistry, impact of airborne dust content on radiation balance, dust–cloud interactions, darkening of snow/ice surfaces, glacial melting, change in emissive surfaces due to glacier retrievals, and many more. Further research needs to refine the sources parameterization methodology, such as including sources distribution annual dynamics and better representation of source emission capacities in order to avoid potential overestimation of emissions, false alarms and underestimation of emissions of other potentially emissive areas.

Based on the presented research, Republic Hydrometeorological Service of Serbia (RHMSS) has established a daily Icelandic dust forecast since April 2018 [83,84]. Forecast products are also shared at the WMO SDS–WAS portal (https://sds-was.aemet.es/news/new-icelandic-dust-forecast, accessed on 25 May 2022).

**Author Contributions:** B.C. and S.N. initiated the research; S.P. and B.C. performed the DREAM model experiments; Ó.A., P.D.-W. and L.L. performed soil data analysis; F.M. and M.R. performed Lidar/Ceilometer data analysis; E.P., A.G. and V.A. performed satellite data analysis; D.C. performed surface PM and air quality data analysis; A.V.V. contributed to model setup, experiments and results analysis; J.N. contributed to the results interpretation. In addition, all authors contributed to interpretations, formulated the conclusion, and reviewed the manuscript. All authors have read and agreed to the published version of the manuscript.

**Funding:** This research received no external funding.

**Institutional Review Board Statement:** Not applicable.

**Informed Consent Statement:** Not applicable.

**Data Availability Statement:** Data available on request due to their robustness and restrictions on public sharing.

**Acknowledgments:** The support for the research and development of the Icelandic dust model is provided by the Republic Hydrometeorological Service of Serbia. This research is partly supported by the EU COST Action CA16202 "International Network to Encourage the Use of Monitoring and Forecasting Dust Products (inDust)". The study of the high latitude dust represents a part of research and operational activities in WMO SDS–WAS (World Meteorological Organization Sand and Dust Storm Warning Advisory and Assessment System, https://sds-was.aemet.es/, accessed on 25 May 2022). Part of the study related to observation (EP) has been supported by the project PANhellenic infrastructure for Atmospheric Composition and climatE change (MIS5021516) which is implemented under the Action Reinforcement of the Research and Innovation Infrastructure, funded by the Operational Programme "Competitiveness, Entrepreneurship and Innovation" (NSRF2014–2020) and co–financed by Greece and the European Union (European Regional Development Fund). The Icelandic Research Fund (Rannis) supported this project with Grants No. 152248–051 and No. 207057–051; partial support the InDust COST project. VA and AG acknowledge the support by the European Research Council (grant no. 725698, D–TECT). The authors gratefully acknowledge the NOAA Air Resources Laboratory (ARL) for the provision of the HYSPLIT transport and dispersion model and/or READY website (https://www.ready.noaa.gov, accessed on 25 May 2022) used in this publication. AVV scientific research was partially funded on the basis of the contract on scientific research funding (between Ministry of Education, Science and Technological Development of RS and Faculty of Agriculture, University of Belgrade) no. 451–03–9/2021–14/200116. The authors gratefully acknowledge Rakul Mortensen, Environment Agency of Faroe Islands, for provision of surface PM measurements. The particle size distribution data collection is supported by the internal program of the Institute of Geology CAS in Prague No. RVO 67985831. Publication of this paper and related research was funded by Czech Science Foundation Grant No. 20–06168Y. We acknowledge the use of imagery from NASA's Worldview application (https://worldview.earthdata.nasa.gov, accessed on 25 May 2022), part of NASA's Earth Observing System Data and Information System (EOSDIS). The CALIPSO data were obtained from the online archive of the ICARE Data and Services center http://www.icare.univ-lille1.fr/archive, accessed on 25 May 2022 (NASA/LARC/SD/ASDC. (2018); ICARE Data Center, 2021) NASA/LARC/SD/ASDC. (2018). CALIPSO Lidar Level 2 Aerosol Profile, V4-20 [Data set]. NASA Langley Atmospheric Science Data Center DAAC, retrieved from https://doi.org/10.5067/CALIOP/CALIPSO/LID_L2_05KMAPRO-STANDARD-V4-20, accessed on 25 May 2022, ICARE Data Center: CALIPSO data, available at: http://www.icare.univ-lille1.fr/, accessed on 2 February 2021. LL scientific research was supproted by OP RDE, MEYS, under the project "Ultra-trace isotope research in social and environmental studies using accelerator mass spectrometry", Reg. No. CZ.02.1.01/0.0/0.0/16_019/0000728.

**Conflicts of Interest:** The authors declare no conflict of interest.

## Appendix A. Particle Size Distribution Observations

The grain size distribution was determined using a laser particle size analyzer CILAS 1190 LD (measurements range from 0.04 to 2500 µm). The measurement was taken after sufficient dispersion was provided by reaction of the sample with KOH for 10 min.

## Appendix B. MODIS AOD

Columnar aerosol observations from the MODerate resolution Imaging Spectroradiometer (MODIS), onboard the Terra and Aqua polar orbit satellites, have been processed for the assessment of the Icelandic dust outbreaks' numerical simulations. More specifically, the aerosol optical depth at 550 nm ($AOD_{550}$), obtained from the latest version (Collection 6.1) of the MODIS retrieval algorithms, has been utilized. As it has been described in [85], the AODs acquired from three retrieval algorithms [86–88], relying on different assumptions based on the underlying surface type, have been merged. At each MODIS swath (Level 2 data; L2), only the most reliable (QA = 3; [89]) AODs are kept, which subsequently are combined and are regridded on a daily basis in order to depict the suspended loads' patterns at fine spatial resolution ($0.1° \times 0.1°$). The MODIS data that have been used for the purposes of the current study are freely available from the Level–1 and Atmosphere Archive & Distribution System (LAADS) Distributed Active Archive Center (DAAC) (https://ladsweb.modaps.eosdis.nasa.gov/, accessed on 25 May 2022).

## Appendix C. CALIOP

High-resolved observations of aerosols, for the vertical assessment of the Icelandic dust outbreaks numerical simulations, have been provided by the Cloud–Aerosol Lidar with Orthogonal Polarization (CALIOP), the elastic backscatter lidar and primary instrument on-board the sun-synchronous, polar-orbit and part of the Afternoon–Train (A–Train) constellation of Earth-observation satellites, Cloud–Aerosol Lidar and Infrared Pathfinder Satellite Observation (CALIPSO) [90]. In the framework of the study, we utilize CALIOP Version 4.2 Level 2 (L2) aerosol and cloud profiles of backscatter coefficient and particulate depolarization ratio at 532 nm [91], in addition with the feature type and aerosol subtype of the detected atmospheric layers [68], along the CALIPSO orbit track. In addition, a methodology originally established in the framework of the Aerosol Research Lidar Network (EARLINET) [92], towards the decoupling of the pure-dust component from the total aerosol load [93], is applied to CALIPSO–CALIOP observations [94]. More specifically, the methodology includes CALIOP L2 and L3 quality assurance procedures [95,96], assumes CALIOP aerosol subtypes "polluted dust", "dusty marine", and "dust" as external mixtures of a pure-dust and a non-dust component, towards the derivation of CALIPSO-based pure-dust extinction coefficient profiles at 532 nm [97].

## Appendix D. Lidar/Ceilometer Observations

It is well known that ceilometers are useful instruments to characterize the vertical structure of the atmosphere in an operational way, although without an appropriate calibration [78,81,98], as they cannot estimate the aerosol attenuated backscatter. Moreover, quantitative estimation of aerosol optical properties may be affected by large uncertainties due to the lower signal-to-noise of this type of instruments. In the absence of ancillary information, from a Raman lidar, a sun-photometer or air mass back trajectories, the typing of aerosol particles may be challenging.

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
