# Peer review of "Fully Dynamic High–Resolution Model for Dispersion of Icelandic Airborne Mineral Dust"

_atmosphere, doi:10.3390/atmos13091345_

Round 1
Reviewer 1 Report
Please see the attached review report.

Reviewer 2 Report
This work established a fully dynamic high-resolution dispersion model for icelandic airborne mineral dust based on the Dust Regional Atmospheric Model (DREAM) by improving dust source specification, particle size distribution, highly erodible areas, variable snow cover, and soil wetness effects. The newly developed model well captured several historical large scale dust storm with robust evidences from satellite remote sensing data and field observation data. The model and the forecast products have made important contribution the relevant scientific community and further evaluation on Icelandic dust. Generally, the manuscript is well organized and well written, with clear logic, detailed method description, sufficient evidences, full discussion, and fluent language, though it is a little long. There are several minor comments or suggestions for the authors to consider.
1. Abstract, the first paragraph, Line 18-28, suggest simplifying this paragraph into several sentence to only demonstrate the significant impacts of dust and the weakness of previous models. Too many descriptions on the background will hide the importance of the current work.
2. Abstract, the third paragraph, Line 40-43, suggesting adding a few important and detailed results on the spatial range, vertical distribution, and temporal variations of the captured dust storms.
3. Line 102 and through the main text and figure titles, pay attention to the subscript for the PM10.
Reviewer 3 Report
General comment
The paper reports a study for development of the coupling of DREAMS model with local emissions of Icelandic dust for high resolution analysis of dispersion of this dust. The topic is interesting and it has elements of novelty. The paper is well written and suitable for the Journal. However, there are some aspects that should be discussed in more detail including performances of this model, see my specific comments. I suggest to consider the paper for publication after a major revision.
Specific comments
Line 22. What do you mean withe excessive atmospheric circulation? Excessive should likely be deleted.
Lines 86-88. Why observed at weather stations? I believe that they are air quality stations. Furthermore it could be useful to mention that the frequency of these dust events is much larger than that observed for African dust events in the Mediterranean basin, see for example the analysis of Conte et al. (Atmospheric Research 233 (2020) 104690).
Lines 104-107. This is not clear. Do you mean that there is an average ratio PM1/PM10 around 0.5?
It would be better to specify that the size distributions of figure 4 are relative to the emissions and used in the model because the size distributions on receptor sites would be different according also to distance from the emissions.
Line 259. What is the meaning of bold KH and KV? In equation (6) they are not bold.
Line 286. What is the laminar air viscosity? It does exist dynamic viscosity or kinematic viscosity.
Line 310. Please remove he repeated word “model”.
Line 378. Use an apex for “-3”. The same for “-1” at line 593.
Line 471. No need for the word exquisitely. The same in line 590.
In my opinion there seems to be an overestimation of ground level PM10 figure 8, 15, 21 and, often, secondary peaks obtained by the model not observed in measurements. This aspects should be discussed in more detail and conclusions lines 637-640 and lines 652-656 should be a little more critical on this outcome.
Round 2
Reviewer 3 Report
Authors revised the paper and improved it. My questions have been reasonably answered and I suggest to accept the paper for publication in the current form.